# Monitoring Means and Results of Biosecurity in Pig Fattening Farms: Systematic Assessment of Measures in Place and Exploration of Biomarkers of Interest

**DOI:** 10.3390/ani12192655

**Published:** 2022-10-03

**Authors:** Annalisa Scollo, Pierre Levallois, Christine Fourichon, Ambra Motta, Alessandro Mannelli, Francesco Lombardo, Paolo Ferrari

**Affiliations:** 1Department of Veterinary Sciences, University of Torino, Grugliasco, 10095 Torino, Italy; 2Oniris, INRAE, BIOEPAR, 44300 Nantes, France; 3CRPA Research Centre for Animal Production, 42121 Reggio Emilia, Italy; 4Swivet Research sas, 42121 Reggio Emilia, Italy

**Keywords:** biosecurity, pig, tailor-made plan, biomarkers, slaughter check

## Abstract

**Simple Summary:**

In recent times, the interest in biosecurity in pig facilities has grown exponentially due to the severe threat to the swine industry worldwide represented by epidemic pathogens and globalization. The outcomes of biosecurity have been monitored through antimicrobial use, and economic and technical performances in pig farms, but limited quantitative data are available regarding animal-based biomarkers over time as output parameters. By means of tailor-made biosecurity protocols, we monitored—over a 12-month period—the biosecurity implementation in swine farms, and described animal-based biomarkers as output parameters. The results shown by the present study suggest that a systematic evaluation of biosecurity is a useful approach to formulate a tailor-made biosecurity plan, and to monitor its implementation; biomarkers could bring insight into the outcomes of biosecurity. Moreover, the description of four farm biosecurity profiles in the present study might orient further investigation in the future on the possible link between biosecurity and lung lesions and scars at slaughter.

**Abstract:**

Limited data are available regarding animal-based biomarkers over time as outcomes of biosecurity in pig farms. The aim of this study was to gain an insight into the biosecurity implementation in a convenience sample of 15 swine herds, and to describe potential biomarkers of interest; inputs from a systematic evaluation of biosecurity implementation were used to develop tailor-made biosecurity protocols monitored over a 12-month period. The farms’ implementation was then described, and animal-based biomarkers were explored as output parameters. A significative biosecurity improvement was observed at the end of the study (*p* = 0.047), in particular in the professional zone (*p* = 0.012). Four clusters of farms were identified for their progress on biosecurity implementation by means of Principal Component Analysis (PCA) and Hierarchical Cluster Analysis (HCA): 4/15 farms improved their biosecurity only in the professional zone, 8/15 showed scarce/null improvement of total biosecurity, 2/15 worsened their biosecurity, and 1/15 greatly improved biosecurity. The farm biosecurity profiles showing an improvement included farms with a reduction in lung lesions and scars at slaughter at the end of the study. The results suggest that a systematic evaluation of biosecurity is a useful approach to formulate tailor-made biosecurity plans and monitor their implementation; biomarkers might bring insight into the outcomes of biosecurity.

## 1. Introduction

Biosecurity, as defined in the literature, is a set of structural, logistical–managerial, and behavioural measures designed to eliminate or reduce the risk of the introduction, establishment and spread of disease-causing agents in a population (or a facility/vehicle, zone or compartment) in order to ensure the protection of public, animal and environmental health (the “One Health” concept) [1]. In recent times, the interest in biosecurity in pig facilities has grown exponentially due to the severe threat to the swine industry worldwide represented by the challenging spread of African swine fever virus across Europe, with the last new outbreaks in Italy in January 2022. This is an example of the importance of biosecurity, as the spread of the virus into new countries leads to devastating and unrecoverable socio-economic losses for the entire swine production sector due to trade restrictions on animals and animal products [2].

The quantitative measure of biosecurity is of the utmost importance in order to identify shortcomings, prioritize improvements and monitor biosecurity variability over time. Benchmark farms should be identified in order to increase biosecurity levels from the local to the national level. [3,4].

Checklists have been adopted in the field to translate information regarding biosecurity into a score for a specific herd, providing an objective, comprehensive and quantitative description of the level of biosecurity that can be used to inform the farmer of possible areas for improvements, and to compare that biosecurity level with other farms/herds (i.e., benchmarking). Having a good insight into differences and similarities in the biosecurity level of pig herds is of great importance for the advice that can be given to a single farm in order to improve its biosecurity status, and also to improve the biosecurity status of pig production in the EU and thereby lower the probability of the introduction and spread of diseases. Moreover, the strict application of preventative biosecurity practices and the separation of animal populations, in order to distinguish animal populations with differentiable health statuses, lead to the recent concept of compartmentalization [5].

However, motivating farmers to invest in new structures and change their daily routines is a well-known challenge [6], and biosecurity measures seem particularly difficult to implement only by legal requirement [7]. Moreover, there is no ‘‘one-plan-fits-all’’; each plan should be tailored to meet the needs of management’s goals and expectations, and problems specific to a production enterprise or geographic region [8].

Recently, some authors described the effects of tailor-made and prevention plans by monitoring antimicrobial use, economic, and technical performances in pig farms. Following interventions, a substantial reduction in antimicrobial use was achieved without a negative impact on the overall farm technical performance and treatment frequency against predefined categories of disease symptoms, with the latter being used as a proxy of disease incidence [9,10,11]. However, limited quantitative data are available in the literature to describe the implementation of biosecurity and to monitor its improvement over time when a tailor-made plan is designed in a farm. Moreover, besides monitoring efforts in the implementation of biosecurity measures (the input or resource-based approach), it could be of interest to monitor the outcomes of the expected prevention using animal-based biomarkers (the output-based approach). The literature describes biomarkers as indicators of a biological process or pathological states which are able to provide information on the current status of the future risk of disease of an individual [12].

The aim of the present study was to gain an insight into the implementation and improvement of biosecurity in a convenience sample of 15 swine herds, and to describe potential biomarkers of interest in these farms; inputs from a systematic evaluation of biosecurity implementation were used to develop tailor-made herd protocols to improve biosecurity, and were monitored over a 12-month period. The farms’ status was then described using some animal-based biomarkers as output parameters.

## 2. Materials and Methods

The study was carried out in the context of the project “Healthy Livestock, Tackling Antimicrobial Resistance through improved livestock Health and Welfare”, from the European Union’s Horizon 2020 research and innovation program. The method proposed in the present study incorporates both input and output parameters in order to assess the risks of pathogen introduction, exposure and spread on modern intensive pig farms. The input parameters are collected by a Biosecurity Risk Analysis Tool (BEAT) regarding the risk of introduction of a pathogen in a farm, the risk of exposure of susceptible animals, and the risk of disease spread within the farm. The output parameters are biomarkers, i.e., animal-based indicators to monitor biosecurity outcomes continuously, and to give an early detection of breaches in biosecurity or biocontainment.

All of the farms (n = 15) included in the study were fattening sites (from 30 kg to a slaughter weight of 170 kg) involved by convenience sampling and randomly selected from a list provided by a contractor under a contract farming agreement, as suggested by de Oliveira Sidinei et al. [13]. This selection procedure was adopted in order to minimize the effect of contractual requirements and different types of contractor–farmer relationships. A list of pig farmers under the same contractual agreement was obtained from the contracting company. The farms were located in North Italy, where heavy pigs are mostly reared for Protected Designation of Origin (PDO) ham production. This area supplies 80% of the national pig production [14].

### 2.1. Development of a Biosecurity Risk Analysis Tool

The questionnaire developed during the present study aimed to describe the complete biosecurity situation in modern intensive pig herds. Questions were asked on each relevant aspect of biosecurity in order to determine whether a preventive measure was applied or whether a specific situation was present or absent. The questionnaire was developed starting from information on general biosecurity procedures in livestock (e.g., equipping the hygiene lock, the hygienic protocol before entering the pigs’ house, etc.) in order to systematically assess biosecurity-related disease risks related to housing and management in pig farms. A structured and comprehensive method was developed based on existing knowledge and expertise, taking into account the FAO 3-zone biosecurity model [15]. A thorough literature study was performed, including existing scoring systems for biosecurity in pig farms. A Biosecurity Risk Analysis Tool (BEAT) for pig farms was delivered to work in Microsoft Excel, including instructions for new users, listing the risks of major diseases of sows, piglets and fattening pigs. The tool was structured by zone, then by the biosecurity objectives to reach (Table 1). As a means of novelty compared to previous methods proposed by the literature, the BEAT included the identification by the auditor of measures not implemented that are considered to be critical in a given farm.

The BEAT was presented and discussed between experts of disease causation (researchers), of disease control (field and official vets of the competent authority), and of farmers (responsible for the daily implementation of biosecurity) within two rounds of consultation in France (French National Institute for Agricultural and Environmental Research—INRAE) and Italy (Research Centre for Animal Production—CRPA). In an interactive process, the structured framework for risk analysis was improved based on the results of the Italian focus group (CRPA) and the consultation of researcher and field vets in France (INRAE).

The final questionnaire included five main sections related to both external (pathogen entry risks in animal husbandry) and internal (pathogen spread between and in animal husbandry departments) biosecurity: the red zone (i.e., outside the farm perimeter, the public zone), the orange zone (i.e., the professional zone in between the pigs’ houses), the green zone (i.e., the pigs’ houses, the herd zone), and the two interfaces between external/professional zones (red/orange) and professional/internal zones (orange/green). Biosecurity and environmental sustainability items were rated on a 4-point scale: a score of 0 was assigned to farms with completely inadequate biosecurity or sustainability practices, and a score of 3 was assigned to those with completely adequate biosecurity or sustainable practices. The scale was divided into 1-point linear increments within this range [16]. When an item was addressed with a yes/no answer, a score of 0/3 was used. The five biosecurity sections of the questionnaire contained 97 items in total (Table 1).

### 2.2. Biomarkers

The biomarkers of interest consisted of animal-based measures, as indirect measurements of animal exposure to pathogens’ presence and spread.

#### 2.2.1. Clinical Scores and Mortality

The animal-based measurements investigated at the farm were the coughing and the sneezing scores related to respiratory disease, and the fecal score to evaluate enteric disease. Clinical evaluation was performed in the fattening farms (n = 15) at the beginning and the end of the growing cycle (30-kg pigs and 170-kg pigs). The clinical evaluation of respiratory disease was obtained by examining, in each farm, at least 100 animals (>50% of the pigs in the same room). The assessor was an expert swine veterinarian. The animals were exhorted to stand, and after 5 minutes of waiting, coughs and sneezing were separately recorded three times for two minutes each. A cough attack (or sneezing) corresponded to 1 cough (or sneeze) [17]. Data for each farm were expressed as the mean percentages of coughing or sneezing animals detected during the 3 collecting times.

The clinical evaluation of enteric disease was performed by means of the Pedersen and Toft grid [18], scoring faeces on a 4-point scale (1 = firm and shaped; 2 = soft and shaped; 3 = loose; 4 = watery). The faeces score was assigned to at least 15 pens per farm by evaluating the worst faeces on the floor, and the mean score was expressed for each farm. The farm mortality rate was calculated on 1-year base.

#### 2.2.2. Slaughter Check

Other animal-based measurements were collected at slaughter: the pluck lesion scores (lungs, pleura, pericardium, and liver) are described in Table 2 [19]. The monitoring of animals was carried out in an abattoir in Emilia Romagna which slaughters around 4500 fatteners per day. Pigs of around 165 kg live body weight and 9 months of age were delivered to the abattoir by trucks in batches of about 135 heads (i.e., minimum 130; maximum 140) derived from the same holding; all of the pigs belonging to the same batch were consecutively slaughtered on the same day [20]. In each batch, about 100 animals (minimum 95; maximum 105) were selected for the pluck evaluation, omitting carcasses at the beginning and end of each batch in order to avoid any risk of accidentally including pigs belonging to the previous or the next batch. The speed of the slaughter line was 480 animals per hour, and the pluck inspection was performed directly during the slaughtering process, from a platform immediately after the evisceration area. The examination of the pluck of each animal was conducted by a veterinarian trained to assign a score for each lesion (Table 2), who alternately worked side by side with the government official veterinarians, using different protocols of evaluation. The examination of the pluck was conducted by the visual inspection and manual palpation of the organs, without any incision. The scores were registered using a voice recorder placed in the upper pocket of the overalls, and were transcribed in an Excel file for analysis during the intervals between work shifts.

The skin lesions of the carcass were also collected. The inspection of the carcass was performed directly during the slaughtering process from a designated position on the line, after scalding and before the de-hairing of the carcass, which was still entire at the time of evaluation. Scores for acute traumatic lesions (scratches) were assigned to the carcass, which was divided into two parts: the “back” region, including the hind legs and the tail, and the “front” region, defined as the remaining area (starting from the loin up to the front limbs, the head and the ears). A 3-point scale system for each of the carcass regions was used in order to easily scan the carcasses during their rapid passage on the dressing line: score 0, up to one scratch or bite; score 1, from two to five scratches or bites; and score 2, more than five scratches or bites, or any wound which penetrates the muscle (similar to the Welfare Quality^®^ Protocol, which differs both for the perimeter of the regions and for the number of scratches per score) [21].

#### 2.2.3. Antimicrobial Use

The defined daily dose/population correction unit (DDDvet/PCU) method proposed by the EMA [22] was adopted in order to quantify the overall antimicrobial use (AMU) per year. The EMA provides standardized daily dosages for each active substance; the PCU was obtained by multiplying the total number of yearly slaughtered pigs by a standard live weight at treatments of 50 kg, as proposed by the EMA.
DDDvet/PCU = total administered active substance (mg)/defined dailydosage (mg/kg/d) × n animals × expected weight at treatment (kg)(1)

### 2.3. Protocol for Data Collection and the Tailor-Made Biosecurity Plan

Prior to the visit, each farm was contacted by telephone and informed of the project in order to obtain consent. All of the contacted farms agreed to the visit. In total, the farms were visited three times during a 12-month study period.

Visit 1: The current/historical biosecurity status was analyzed through the BEAT application in order to assess the implementation of biosecurity measures, and biomarkers were sampled and measured. After the identification of critical issues related to biosecurity in the BEAT, a tailor-made biosecurity plan was planned following suggestions from Donaldson [23]. A final tailored biosecurity plan was written and developed by the farm owner or manager in collaboration with the farm’s attending veterinarian, and with input from farm personnel to ensure that the biosecurity plan could realistically be implemented by the farm personnel.

Visit 2: After 6 months, an in-progress follow-up of compliance with the biosecurity plans was checked. The written plan was modified or updated, as suggested by Donaldson [23].

Visit 3: After 12 months from visit 1, the biosecurity status was analyzed again through the BEAT application, in order to check the follow-up of the compliance with biosecurity plans. The biomarkers were sampled and measured for the second time.

The farm visits were carried out by a trained veterinarian with professional experience in pig production. Strict biosecurity measures were taken at every farm visit. All of the pig farms were visited by the same investigator, such that interviewer bias could be minimized as much as possible, and inter-farm comparability could be ensured. The farmers were interviewed face-to-face by using the BEAT, prior to check the farm, as suggested by Dewulf and Immerseel [3]. After the interview, farm inspection was performed to allow a comparison of the answers given by the farmer and the present situation at the site. A farm visit took one hour on average. If the given answers to the questionnaire did not match the reality which resulted from the farm inspection, the farmer was notified, and the given answer was changed. A maximum of two farms were interviewed per day. The interviews lasted 1.5 h on average.

### 2.4. Data Analysis

The biosecurity scores were calculated according to the method suggested by Diana et al. [24], after partial modification. A value for biosecurity was calculated for each of the five zones by summing the score of those items belonging to each zone. All of these values were then expressed as a percentage from 0 to 100, where zero indicated a poor status (i.e., a lack of biosecurity measures) and 100 indicated a good status (i.e., the full application of the biosecurity measures).

Statistical analysis was performed by XLSTAT 2022.2.1 (Addinson, TX, USA, 2022). Principal Component Analysis (PCA) and Hierarchical Cluster Analysis (HCA) were performed in order to describe the inter-farm variability of the evolution of the five biosecurity score zones over 12 months (active variables). The evolution was expressed as the difference for each parameter between visit 3 and visit 1 (i.e., Δ = visit 3 − visit 1). The evolution of the farm biomarkers and the total biosecurity score (the mean of 5 zones) were included as supplementary variables, meaning that they did not influence the analysis but were used to interpret the results. Factors with eigenvalues equal to or greater than 1.0 (Kaiser criterion) were retained [25]. The HCA allowed us to aggregate farms sharing similar biosecurity score evolutions [13,26]. In the description of the clusters, only variables with a square cosine >0.2 were used, as these were considered the most representative [27].

A further descriptive analysis was carried out on data collected in the first and the last visit, and on the difference over time in the five zones’ biosecurity scores. The Kendall Tau test was applied in order to investigate correlations between the variables in both visit 1 and 3. Statistical significance was set at *p*-values (*p*) < 0.05. The pre- and post- intervention plan results were analyzed using the Wilcoxon signed-rank test to identify the changes of mean scores over time [28].

**Table 2 animals-12-02655-t002:** The scoring system used for pluck lesion evaluation at slaughter in Italian heavy pigs. Each batch comprised a group of about 135 (minimum 130; maximum 140) pigs from the same holding that were slaughtered on the same day. Lesions were scored on around 100 pigs in each batch.

Lesions	Scale	Description
Lungs		
Lung score (Madec score)	0–24	Pneumonic lesions (enzootic pneumonia-like, often due to *Mycoplasma hyopneumoniae*: purple to grey rubbery consolidation, increased firmness, failure to collapse and marked edema) were scored according to Madec’s grid [29]. Each lobe, except the accessory lobe, was scored from 0 to 4, to give a maximum possible total score of 24.
Absence of lesions	0–1	Lungs in which all the lobes, except the accessory one, received score 0.
Severe lesions	0–1	Lungs with a Madec score ≥5/24.
Scars	0–1	Presence of recovered enzootic pneumonia-like lesions, with thickened interlobular purple to grey (depending on the age) connective tissue which appears as retracted tissue.
Abscesses	0–1	Presence of at least one abscess in the lungs.
Consolidations	0–1	Pneumonic lesions complicated by secondary bacterial pathogens (e.g., *Pasteurella* spp., *Bordetella* spp.), firmer and heavier than enzootic pneumonia-like lesions. In the case of a cut surface, lesion was mottled by arborized clusters of gray-to-white exudate-distended alveoli, and mucopurulent exudate could be expressed from the airways [30].
Lobular/chessboard pattern lesions	0–1	Presence of scattered multifocal spots of purple to grey discoloration indicative of probable co-existence of viruses (Porcine Reproductive and Respiratory Virus, Porcine Circovirus, Influenza Virus) and/or *Mycoplasma* spp. or foreign body (e.g., dust/particulate matter) [31].
Pleura		
Pleura score(SPES score)	0–4	SPES grid [32]. 0: Absence of pleural lesions; 1: Cranioventral pleuritis and/or pleural adherence between lobes or at ventral border of lobes; 2: Dorsocaudal unilateral focal pleuritis; 3: Bilateral pleuritis of type 2 or extended unilateral pleuritis (at least 1/3 of one diaphragmatic lobe); 4: Severely extended bilateral pleuritis (at least 1/3 of both diaphragmatic lobes). Most probable etiology: *Actinobacillus pleuropneumoniae*, *Heamophilus parasuis*, *Pasteurella* spp., *Bordetella* spp., *Mycoplasma hyorhinis*.
Severe lesions	0–1	Pleura with a SPES score ≥3.
Sequestra	0–1	Presence of at least one sequestra in the lungs (acute: firm, rubbery and mottled dark red purple to lighter white areas with abundant fibrin, and hemorrhagic, necrotic parenchyma; or chronic: resolution of non-necrotic areas from acute infections results in remaining cavitated necrotic foci that are surrounded by scar tissue). Often associated with *Actinobacillus pleuropneumoniae* infection [33].
Actinobacillus pleuropneumoniae index (APP index)	0–4	Frequency of pleuritis lesions with a SPES score ≥2 in a batch mean pleuritis lesion score of animals with SPES ≥2. The APP index ranges from 0 (no animal in the batch showing dorsocaudal pleuritis) to 4 (all animals with severely extended bilateral dorsocaudal pleuritis) [20].
Liver		
Liver score	1–3	Scoring based on the number of milk spot lesions due to *Ascaris suum* presence and their migration. 1: no lesions or less than 4 lesions; 2: from 4 to 10 lesions; 3: more than 10 lesions.
Severe lesions	0–1	Livers with a score 3.
Total lesions	0–1	Livers with a score ≥2.

## 3. Results

The average number of pigs reared in the visited farms was 2203 ± 1949 (minimum = 300; maximum = 6300). A descriptive analysis of the farms in visit 1 and 3 is reported in Table 3, whereas a graphical illustration of the biosecurity score for each farm in visit 1 and its post-intervention score are given in Figure 1.The average improvement post-intervention of the total biosecurity score was 1.1 ± 2.0% (from 55.7 ± 8.7 to 56.8 ± 9.4%, *p* = 0.047), with 9 farms that improved their total biosecurity score (60% of farms), one farm that worsened its score (6.7%) and 5 farms that did not make any intervention (33.3%). The only zone with a significative improvement between the start and end of the experimental period was the professional zone (mean = 4.8 ± 6.8%; minimum = 0.0%; maximum = 22.5%; *p* = 0.012): eight farms improved the biosecurity score related to this zone (53.3%), and seven did not make any intervention (46.7%). In particular, the practices in the professional zone with the greatest improvement were contamination by manure (+9.2 ± 12.9%), pathogen persistence (+6.7 ± 14.8%), and contamination by staff storing dead animals (+8.3 ± 17.5%). The evolution related to the public zone was a worsening of −0.4 ± 3.8% (minimum = −10.4%; maximum = 6.3%); that related to the transition zone between the public and professional zone was 0.7 ± 2.1% (minimum = −3.1%; maximum = 4.2%); that related to the transition zone between the professional and herd zone was 0.4 ± 1.1% (minimum = 0.0%; maximum = 3.9%); and that related to the herd zone was 0.0 ± 1.3% (minimum = −2.8%; maximum = 2.8%).

A Wilcoxon signed-rank test identified significant changes of mean values over time in the mean percentage of animals showing skin lesions at slaughter in the posterior region of the carcass (16.5 ± 10.9% vs. 2.0 ± 1.9%; *p*-value 0.027). A trend was shown for the percentage of sneezing animals before slaughter (around 170 kg b.w.; 1.0 ± 1.1% vs. 0.5 ± 0.5%; *p*-value = 0.064) and for the AMU (22.7 ± 11.0 vs. 15.2 ± 13.6 DDDvet/PCU; *p*-value = 0.088).

The PCA (Figure 2) and HCA performed on the evolution of both biosecurity scores (as active variables) and biomarkers (as supplementary variables) resulted in the identification of four clusters (A, B, C, D) of farms. The main characteristics defining the clusters of farms are presented in Table 4, and a description of the clusters is provided in the text below.

From the cluster analysis, the first two factors (eigenvalues >1.0) synthesized by the PCA accounted for 83.8% of the variability. Variables with a squared cosine >0.2 were considered to contribute significantly to the two factors, and are reported in bold in Table 4.

Cluster A: Farms that limited the improvement to their biosecurity score only to the professional zone. Four farms (26.7%) were characterised by efforts to increase by +7.5% the biosecurity score in the professional zone but limited or null improvements of the other zones. In particular, “pathogen persistence” and “contamination by staff storing dead animals” accounted for most of the improvement (+18.7% and +9.4%, respectively). Among the biomarkers, a reduction in lung scars at slaughter was observed (−7.0%).

Cluster B: Farms with scarce/null improvement of the total biosecurity score. This cluster represents most of the farms involved in the study (8 farms, 53.3%), which showed a very limited improvement in the total biosecurity score after the 1-year study, even if they succeeded in slightly improving the risk factors of the public zone “dead animals” and “external vehicles” (+3.1% and +1.6%, respectively) more than the other farms. No obvious changes in biomarkers were observed.

Cluster C: Farms that worsened their total biosecurity score. Two farms belonged to this cluster. In particular, even if they improved the biosecurity in the professional zone, they lost a relatively important percentage of their score in the public zone. The risk factors that showed the most representative loss were “dead animals” (−20.8%), “animal contact with contaminated premises” (−18.7%), “external vehicles” (−9.4%), and “unnecessary access” (−9.4%). No obvious output from the biomarkers might be described, excepting a slight increase in the lung lesions score.

Cluster D: The farm that greatly improved the total biosecurity score. Only one farm excelled in improving the biosecurity score, especially with the intervention in the professional zone (+22.5%). A reduction in lung lesions and lung scars was observed at slaughter.

No relationships were found according to the Kendall Tau test between biosecurity scores and biomarkers, except for a correlation of lung lesion scores vs. lung scars (correlation = 0.451, *p*-value = 0.015), and lung lesion scores vs. pleural (SPES) scores (correlation = 0.433, *p*-value = 0.020).

## 4. Discussion

This study describes the use of a novel biosecurity risk analysis tool, called BEAT, to identify strengths and weaknesses on pig farms based on existing knowledge of risk factors, previously existing biosecurity checklists, and the FAO 3-zone biosecurity model [15]. The combination of the conceptual approaches suggested by the existing checklists and the FAO 3-zone biosecurity model is expected to provide the assessor and the farmer with more detailed insight into farm facilities and management in each area, which is the basis for promoting a more careful risk analysis and more precise identification of mitigation measures. The aim was to gain an insight into the biosecurity of a convenience sample of swine herds. The BEAT was applied and tested in a sample of 15 Italian fattening pig farms, providing an assessment of the initial implementation rates and evolution over 12 months of observable biosecurity measures in the different zones of the farms, resulting into four farm profiles based on the improvement of farm biosecurity. For each farm profile, a description of the biomarker outputs during the study was provided.

The average biosecurity improvement during the study was statistically significant, although not all of the farms implemented new biosecurity measures over time. Implementation after the adoption of a tailor-made biosecurity plan is more likely to be achieved, as it is written and developed by the farm owner or manager in collaboration with the farm’s attending veterinarian, and with input from farm personnel [23]. This approach ensures that the biosecurity plan can realistically be implemented by the farm personnel, establishing partnerships to identify “best practices” for stock management that includes biosecurity practice. The farms that failed in the implementation of biosecurity measures probably reflect the scarce motivation to invest in new structures and change their daily routines reported by other authors [6], lead by a poor perception of benefits. Knowledge is often seen as key to changing behaviour; if individuals do not know the impacts of their actions, then they cannot be expected to change their attitude toward a certain challenge [34]. For example, farmers are likely to be more motivated to implement biosecurity measures if such measures can be expected to be beneficial for their farm performance [35,36,37].

The professional zone (i.e., between the animal houses) was the most improved zone during the 12 months of the study. Among the measures belonging to this zone, the farmers’ approach to manure management for the prevention of contamination in the professional area was improved by 9.2%. Manure can be an important source of many infections (swine dysentery, classical swine fever, foot and mouth disease), as well as *Escherichia coli*, porcine reproductive and respiratory syndrome virus, *S. suis*, *Salmonella* spp., and so on. The risk is particularly high if the manure comes from other pig farms [38]. The equipment used for spreading slurry should be, whenever possible, dedicated to the premises, as it becomes a significant threat to the biosecurity of the farm if that equipment has been used on other farms with lower health statuses [39,40]. As there is considerable diversity in farm designs and flooring, which have an influence on how the waste from the pigs’ accommodation can be removed and stored [41], a tailor-made plan is again suggested to better improve the manure management. Tailor-made biosecurity plans also improved the prevention of contamination from dead pigs due to poor management in storing them (+8.3%). Both manure and carcass management are operations that require some level of contact with the farm, and the most adequate approach for those operations is a proper design of the farm. The activities requiring such contact need to be located, as far as possible, in the external perimeter, with no need to enter the farm. However, and also when containers for dead animals are located in a correct position, truck drivers or other personnel involved in carcass management should never be allowed to enter into contact with the animals and the clean areas of the herd zone. Kim et al. [42] showed the importance of this measure. They examined the transmission of porcine epidemic diarrhoea virus under low and high biosecurity measures, and observed that the clothes and boots of personnel exposed to infected animals easily became contaminated with amounts of virus which were likely causing transmission, particularly for boots and overalls. Small amounts of contaminated faeces on boots used by the staff in charge of storing dead animals could be sufficient to infect a farm [4]. The last improved item was the prevention of the persistence of pathogens in the professional zone throughout washable surfaces and floors, and debris removal (+6.7%). Regarding hygienic measures, the most basic element is cleaning the environment. However, the removal of roughness from the floor and surfaces is also critical in order to ensure the good prevention of pathogen persistence in the professional zone. Debris around the barns is also considered a risk, as it may hide rodents that can carry numerous pathogens that affect pigs, such as some *Salmonella serovars*, *Leptospira*, *Yersinia pseudotuberculosis*, *Toxoplasma gondii*, *Campylobacter* spp., *Brachyspira* spp., *Lawsonia intracellularis*, and the encephalomyocarditis virus [4]. Debris also allow the breeding of insects and rodents which will attract wild birds and reptiles, which are potential vectors for infectious diseases close to farm facilities (e.g., from infected carcasses). In fact, those rodents and wild birds may have access to and contaminate pig feed and the materials [43].

The improvement of the professional zone, which represents a part of the internal biosecurity of the farm, might reflect the historical lack of organized internal areas to control diseases, or the barns’ heterogeneity due to the growing of the size of many farms during the last few years by adding new buildings to older, but still functional, facilities, as observed by da Costa et al. [44]. The authors suggested that this is why farmers did not valorize the pertinence of internal biosecurity in the past. In fact, the biosecurity score of the professional zone at the beginning of the study was lower than the score in the public and in the herd zone. Moreover, Casal et al. [35] stated that farmers are likely to implement the biosecurity measures they perceive as important. The awareness of biosecurity has traditionally been focused on external biosecurity by farmers to avoid non-endemic diseases entering their farms [37]. However, the results of the present study might be in agreement with da Costa et al. [44], who suggested that, in recent years, the key importance of internal biosecurity practices to reduce disease and improve profitability has resurged and gained new strength.

This study distinguished four groups of farms according to their level of improvement in biosecurity. Four farms (26.7%) limited the improvement to their biosecurity score to the professional zone only (cluster A). In this cluster, a reduction in lung scars at slaughter was described (−7.0%). The removal of manure/dead animals and the cleaning and disinfection procedures were the main improvements in the professional zone in this cluster; they were reported in the literature as measures which are likely to improve some productive parameters (e.g., daily weight gain and antimicrobial usage), probably due to the reduction of the risk of pathogens entering into or spreading within a herd [37,45]. However, these authors described a high overall biosecurity status in the farms with these improved biomarkers, unlike the present study. In cluster A, the observation of lung scars at the abattoir can help to formulate a hypothesis about what might have happened in the respiratory tract earlier in the life of the animals, in the middle growth cycle. The significant occurrence of pneumonia scar tissue in lungs from pigs might help to pinpoint in time the spreading of an infectious agent [19]. Caswell and Williams [46] and Maes et al. [47] reported that, whilst a fresh lung lesion is visible on the pluck by two weeks after infection, and for at least two months, if the lesions occurred earlier in time, the pluck would show only a scar. The description of cluster A suggests that other studies are needed in order to investigate the hypothesis that the improvement of the professional zone might reduce the spread of pathogens during the first stages of growth of the animals, reducing healing lung tissue at slaughter.

Several farms (53.3%) failed in improving the biosecurity score related to all zones (cluster B). In these farms, a change in all of the biomarkers was also not described, being in agreement with the hypothesis described above.

In recent years, some studies explored the factors influencing decision-making for pig farmers, as well as their attitude towards biosecurity [35,36]. Most authors agreed on the evidence that biosecurity measures are sometimes inconsistently applied on commercial farms, despite most farmers being concerned about biosecurity and aware of its importance in preventing and controlling diseases. Several authors declared that non-compliance with biosecurity measures is thought to be related to the poor training of farm personnel, and to a lack of communication between the personnel working on the farm and the technical services [35], particularly as it relates to the understanding of the meaning of each measure with regard to the transmission of diseases. However, in the present study, it was supposed that the application of a tailor-made biosecurity plan fulfilled the lack of communication between farmers and biosecurity advisors, relocating the cause of absent improvement in other factors. Several authors have suggested that the risk perception of a disease and its consequences on the farm is the main factor leading to the application of biosecurity measures. The greater application of biosecurity measures has been observed after outbreaks of diseases such as Porcine reproductive and respiratory syndrome [7,48] or influenza [49], as well as in densely populated areas of pigs, probably due to a higher perception of the transmission risk between neighbours [50]. It is important to highlight that the present study was entirely performed before the African swine fever outbreak in Italy, which confirmed the virus in a wild boar in the province of Alessandria (Piedmont region, northwestern Italy) on 7 January 2022, followed by several other notices of positivity from wild boars in the Italian peninsula. Probably, a new and similar data collection in the near future could result in a reduced number of farms belonging to this cluster B, as the level of biosecurity improvement would be expected to be higher after an epidemic of infectious diseases [51].

The third cluster (C) was represented by farms that worsened their total biosecurity score, in particular relating to the public zone. Only a slight increase in lung lesion scores was shown in these farms. This condition might reflect one of the main problems for such long-term programs: if they are effective, the result will be that no new diseases will enter the farm, or the spread of existing ones among the different barns will be reduced. If it is not effective, the negative results might be notified after an undetermined time frame, as the entry of new diseases through a faulty public zone is quite unpredictable. In other words, if the program is successful, nothing will happen, as well as in case of unsuccessful program until an unpredictable outbreak. This will give a false perception of the risk, and might lead to the relaxation of the implementation of biosecurity practices in the first case, or to the feelings of uselessness of the interventions in the latter which, in turn, could increase the probability of disease introduction or transmission [4].

The opposite results for cluster D were unfortunately represented by only one farm that greatly improved the total biosecurity score. Even if a reduction in lung lesions and scars at slaughter might confirm the dissertation detailed above, the presence of only one farm in this cluster limits the possibility of other discussion.

Despite the clusters, the percentage of posterior scratches observed at slaughter on the pigs’ skin decreased. Probably, the improvement of this parameter was due to a general increased attention to welfare procedures during transport adopted by the farms and the slaughterhouses, led by the recent Italian governmental pressure on that topic (Classyfarm program, Italy). A similar motivation is supposed for the generalized decreasing trend in antimicrobial use, which confounded the possible link with biosecurity measures previously observed by other authors [37].

The results from the present study might support the hypothesis of other authors that the development of a tailor-made program from a close communication between farmers and veterinarians might increase the availability and credibility of information. For example, different studies have shown that veterinarians are the source of information in which farmers place greater confidence when animal health and biosecurity are dealt with [52,53]. This collaboration might also reduce the risk of failure, due to the mandatory approach of some biosecurity plans requested by institutions in the case of challenging epidemiological statuses (e.g., African swine fever outbreaks in a country), due to the low confidence of farmers in government institutions. Moreover, tailor-made biosecurity plans allow producers to be more involved and responsible for the application of biosecurity measures, as many of them believe that biosecurity should be a matter solely for health organizations, particularly when the measures are intended for zoonosis control, or are applied by international legal or market pressures [35,54]. The tailor-made plan would avoid the perception that various guides, manuals and materials developed by governments and institutions to persuade producers and veterinarians on why and how to apply biosecurity measures have little real impact, as producers think that those recommendations are irrelevant or impractical, even for those who have had disease outbreaks or may receive financial support [4].

A limitation of the study is the restricted number of farms involved, although this does not affect the aim of the paper to gain an insight into the implementation of biosecurity and to describe potential biomarkers of interest.

## 5. Conclusions

The application of a tailor-made protocol on biosecurity and the monitoring of biosecurity implementation by biomarkers over time is suggested as a novel and promising approach. In order to improve the application of biosecurity measures, farmer’s awareness should be constantly increased, probably using participatory methods, as suggested by Alarcón et al. [4]. The systematic evaluation of biosecurity is confirmed as a useful approach to formulate a tailor-made biosecurity plan, and to monitor its implementation. Biomarkers could bring important insight into the outcomes of biosecurity, and the description of four farm biosecurity profiles in the present study might orient further investigation in the future on the possible link between biosecurity and lung lesions and scars at slaughter.

## Figures and Tables

**Figure 1 animals-12-02655-f001:**
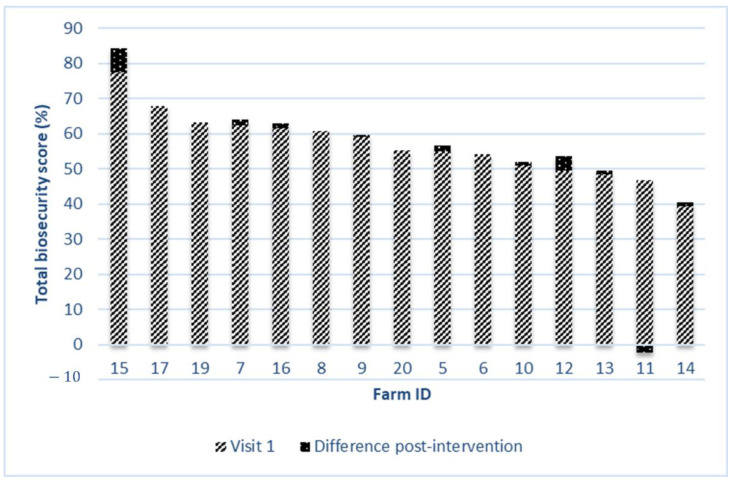
The total biosecurity scores from 15 different fattening farms involved in the study, and their difference post-intervention.

**Figure 2 animals-12-02655-f002:**
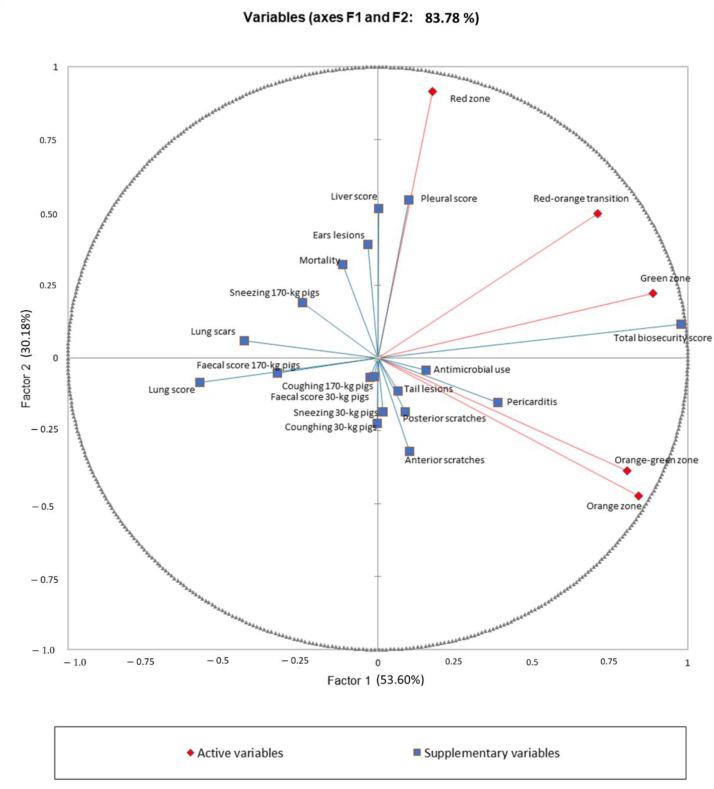
Correlation circle of the PCA analysis. Active variables are changes in biosecurity scores in each zone after 12 months, and are represented in red; red-public zone, red/orange-public/professional transition, orange-professional zone, orange/green-professional-herd transition, and green-herd zone. Supplementary variables are represented in blue.

**Table 1 animals-12-02655-t001:** Risk factors, their objective and number of questions in each of the five zones of the biosecurity checklist.

Zone	Risk Factor	Objective	Items (*n*)
Public	Neighbourhood activities	Awareness of at-risk situation due to neighbourhood	5
	External vehicles	Maintain in the public zone vehicles and persons with no necessary access to the professional zone	4
	Dead animals	Reduce the load of pathogens associated with elimination of dead animals	3
Public/professional transition	Contamination from truck and visitors	Prevent contamination of the professional zone by trucks and visitors	7
	Contamination by wildlife	Prevent contamination of the professional zone by wildlife	1
	Contamination by staff in charge of elimination of dead animals	Prevent contamination by staff in charge of elimination of dead animals in the public zone	4
	Staff and visitors	Prevent introduction of diseases by staff and visitors entering the farm	8
	Unnecessary access	No unnecessary access to the professional zone	4
Professional	Contamination by wildlife	Prevent contamination of the professional zone by wildlife	2
	Contamination by manure	Prevent contamination by the manure	2
	Pathogen persistence	Prevent persistence of pathogens in the professional zone by washing procedures and debris removal	2
	Contamination by staff storing dead animals	Prevent contamination by staff in charge of storing dead animals in the professional zone	6
Professional/herd transition	Pathogens from animals	Prevent of pathogens from animals introduced into the herd	3
	Pathogens from other purchases	Prevent introduction of pathogens by other purchases	2
	Pathogens from shared equipment	Prevent introduction of pathogens by shared equipment entering the farm	2
	Pathogens from staff/visitors	Prevent introduction of pathogens by staff/visitors	8
	Unnecessary access	No unnecessary access to the livestock zone	4
Herd	Animal contact between age groups	Prevent transmission of pathogens between age groups by animal contacts	2
	Animal contact with contaminated premises	Prevent transmission of pathogens between age groups by premises	2
	Animal contact with contaminated staff	Prevent transmission of pathogens between age groups by staff	4
	Animal contact with contaminated materials	Prevent transmission of pathogens between animals by materials and intervention	5
	High load of pathogens	Reduce risk of exposure to high loads of pathogens	4
	Heterogeneous herd immunity	Reduce at-risk situations due to heterogeneous herd immunity	5
	Contaminated feed or water or enrichment material	Prevent contaminated feed or water or enrichment material	8

**Table 3 animals-12-02655-t003:** Descriptive analysis of the farms (n = 15) at the beginning and at the end of the study: biosecurity scores for each zone, clinical scores, mortality per year, slaughter checks, and antibiotic consumption.

	First VisitMean ± SD (Minimum-Maximum)	Third VisitMean ± SD (Minimum-Maximum)	*p*-Value
Biosecurity scores			
Public zone (%)	61.0 ± 11.2 (43.7–87.5)	60.5 ± 10.2 (47.9–87.5)	ns
Public/professional transition (%)	54.2 ± 16.6 (25.0–80.2)	54.9 ±17.7 (25.0–84.4)	ns
Professional zone (%)	56.2 ± 13.3 (35.0–85.0)	61.0 ± 14.8 (37.5–92.5)	0.012
Professional/herd transition (%)	37.6 ± 7.6 (27.6–48.7)	38.0 ± 7.7 (27.6–48.7)	ns
Herd zone (%)	69.6 ± 7.0 (58.3–81.5)	69.7 ± 7.5 (58.3–81.5)	ns
Total biosecurity score (%)	55.7 ± 8.7 (38.7–70.9)	56.8 ± 9.4 (39.5–77.5)	0.047
Clinical scores			
Coughing (%)			
30-kg pigs	1.1 ± 1.7 (0.0–6.7)	1.4 ± 1.0 (0.0–3.1)	ns
170-kg pigs	0.7 ± 0.8 (0.1–3.1)	0.5 ± 0.5 (0.0–1.2)	ns
Sneezing (%)			
30-kg pigs	1.5 ± 1.3 (0.2–4.9)	1.0 ± 0.8 (0.0–2.7)	ns
170-kg pigs	1.0 ± 1.1 (0.0–3.8)	0.5 ± 0.5 (0.0–0.4)	ns ^1^
Faeces score (1–4)			
30-kg pigs	1.6 ± 0.4 (1.1–2.4)	1.6 ± 0.0 (1.1–2.4)	ns
170-kg pigs	1.0 ± 0.0 (1.0–1.1)	1.0 ± 0.0 (1.0–1.1)	ns
Mortality (%)	4.8 ± 1.7 (2.7–8.8)	4.9 ± 1.5 (2.5–7.8)	ns
Slaughter checks			
Pluck lesions			
Lung score	0.5 ± 0.3 (0.0–1.0)	0.6 ± 0.4 (0.1–1.3)	ns
Lung scars (%)	8.7 ± 5.3 (0.0–18.7)	6.6 ± 3.1 (3.1–13.6)	ns
Pleural score	0.9 ± 0.3 (0.3–1.5)	0.7 ± 0.5 (0.2–1.5)	ns
Liver score	1.2 ± 0.1 (1.0–1.4)	1.3 ± 0.2 (1.1–1.6)	ns
Pericarditis (%)	7.4 ± 5.6 (0.0 16.7)	4.0 ± 3.6 (0.0–13.8)	ns
Skin lesions			
Ear lesions (%)	2.5 ± 2.9 (0.0–7.8)	1.0 ± 2.5 (0.0–6.7)	ns
Tail lesions (%)	6.3 ± 10.0 (0.0–27.4)	1.0 ± 1.1 (0.0–2.8)	ns
Posterior scratches (%)	16.5 ± 10.9 (3.9–40.3)	2.0 ± 1.9 (0.0–5.0)	0.027
Anterior scratches (%)	19.6 ± 15.2 (0.0–58.8)	11.7 ± 7.1 (0.2–20.4)	ns
AMU (DDDvet/PCU)	22.7 ± 11.0 (2.8–40.8)	15.2 ± 13.6 (1.1–41.2)	ns ^2^

^1^ Presence of a trend: *p*-value = 0.064. ^2^ Presence of a trend: *p*-value = 0.088.

**Table 4 animals-12-02655-t004:** Changes in biosecurity scores after 12 months and biomarker evolution in the four clusters identified with the HCA. Values in bold correspond to variables with a squared cosine >0.2.

Item	Cluster A	Cluster B	Cluster C	Cluster D
N. farms	4	8	2	1
**Active variables**				
Biosecurity scores				
Public zone (%)	**0.0 ± 0.0**	**1.3 ± 2.5**	**−8.3 ± 2.9**	**0.0 ± 0.0**
Public/professional transition (%)	**2.8 ± 1.6**	**0.0 ± 0.0**	**−2.6 ± 0.7**	**4.1 ± 0.0**
Professional zone (%)	**7.5 ± 5.4**	**0.3 ± 0.9**	**8.7 ± 5.3**	**22.5 ± 0.0**
Professional/herd transition (%)	**0.0 ± 0.0**	**0.0 ± 0.0**	**0.6 ± 0.9**	**3.9 ± 0.0**
Herd zone (%)	**0.7 ± 1.3**	**−0.1 ± 0.3**	**−1.4 ± 2.0**	**2.8 ± 0.0**
**Supplementary Variables**				
Biosecurity scores				
Total biosecurity score (%)	**2.2 ± 1.2**	**0.3 ± 0.4**	**−0.6 ± 2.3**	**6.7 ± 0.0**
Clinical scores				
Coughing (%)				
30-kg pigs	0.0 ± 0.6	−0.3 ± 2.0	0.9 ± 0.5	0.7 ± 0.0
170-kg pigs	−0.4 ± 0.7	−0.1 ± 0.6	0.2 ± 0.6	0.0 ± 0.0
Sneezing (%)				
30-kg pigs	−0.7 ± 1.4	−0.4 ± 1.4	0.0 ± 0.1	−0.5 ± 0.0
170-kg pigs	−0.4 ± 1.3	−0.4 ± 0.6	−0.2 ± 0.8	−1.7 ± 0.0
Faeces score (1–4)				
30-kg pigs	0.0 ± 0.3	0.0 ± 0.6	0.0 ± 0.0	0.0 ± 0.0
170-kg pigs	0.0 ± 0.0	0.0 ± 0.0	0.0 ± 0.1	−0.1 ± 0.0
Mortality (%)	−1.1 ± 1.7	0.5 ± 2.5	−1.2 ± 1.7	−0.1 ± 0.0
Slaughter checks				
Pluck lesions				
Lung score	**−0.2 ± 0.4**	**0.2 ± 4.5**	**0.3 ± 1.4**	**−0.5 ± 0.0**
Lung scars (%)	**−7.0 ± 3.2**	**0.2 ± 4.7**	**−3.7 ± 15.1**	**−5.4 ± 0.0**
Pleural score	−0.2 ± 0.4	0.1 ± 0.3	−0.8 ± 0.5	0.2 ± 0.0
Liver score	0.3 ± 0.2	0.1 ± 0.0	−0.1 ± 0.0	0.0 ± 0.0
Pericarditis (%)	−5.0 ± 6.4	−6.1 ± 7.1	−2.8 ±9.7	11.7 ± 0.0
Skin lesions				
Ear lesions (%)	0.0 ± 0.0	−1.7 ± 3.8	−6.2 ± 3.6	−2.2 ± 0.0
Tail lesions (%)	2.1 ± 0.5	−7.4 ± 13.4	0.6 ± 0.4	−4.5 ± 0.0
Posterior scratches (%)	−2.8 ± 8.3	−13.9 ± 8.6	−6.2 ± 8.3	−10.8 ± 0.0
Anterior scratches (%)	−0.9 ± 22.6	−21.4 ± 22.3	12.5 ± 22.4	−12.3 ± 0.0
AMU (DDDvet/PCU)	3.0 ± 20.3	−12.8 ± 10.8	−3.7 ± 13.8	−13.9 ± 0.0

## Data Availability

Not applicable.

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
