# Peer review of "Monitoring Means and Results of Biosecurity in Pig Fattening Farms: Systematic Assessment of Measures in Place and Exploration of Biomarkers of Interest"

_animals, 2022, doi:10.3390/ani12192655_

Round 1

Reviewer 1 Report

A brief summary

Biosecurity in pig farming is of outmost importance for animal health and welfare as well as for food safety, food security and public health. In this study, tailor-made biosecurity protocols were developed and pig farm biosecurity was assessed using clinical data collected at the farms and postmortem findings at slaughter used as biomarkers. Generally, the use of tailor-made biosecurity protocols resulted in increased biosecurity, however this varied among farms and some of them showed no improvement or even worsened their biosecurity. This is a comprehensive study that is important for improving pig farm biosecurity, and in particular for development of farm biosecurity assessment methods.    

 General concept comments

The study is well designed and performed, including statistical analysis and presentation of data. It is well-written manuscript throughout. I have identified one weakness, which is the Conclusions section. Line 493-515.

Conclusions should be supported by the results of the study. However, in the current Conclusions, the discussed topics cover collaboration between veterinary services and farmers and effect of biosecurity tailor-made biosecurity plans on farmers’ awareness, and includes a number of references. These aspects were not investigated and not discussed in the manuscript and are, thus, not conclusions from this particular study. I therefore suggest rewriting the Conclusions focusing on the present findings as stated in the title and the aims of this study. Both systematic assessment of biosecurity and biosecurity biomarkers should preferably be included.

 I agree that the aforementioned aspects of possible consequences of future use of the tailor-made biosecurity plans are relevant and may be mentioned at the end of Conclusions, however without extensive discussion and avoiding references.    

According to the journal policy: “In the text, reference numbers should be placed in square brackets [ ]”. In the Conclusion, the authors used a different style. Please correct (if decided to use references).

Another general comment concerns Biosecurity Risk Analysis Tool (line 113-151). It would be very useful for an interested reader to be able to see the entire Biosecurity Risk Analysis Tool. I could be included, for example, as Supplementary Materials. However, I understand that this may not be possible for various reasons.

Specific comments

Line 469-478. It is somewhat difficult for me to follow the reasoning. See below my suggestions highlighted.  You might want to rephrase this in some other way.  

 “This condition might reflect one of the main problems for such long-term programs: if they are effective, the result will be that the entry of new diseases will not be seen, or the spread of existing ones will not be reduced. If not effective, the negative results might be notified after an indetermined undetermined time frame, as the entry of new diseases through a faulty public zone is quite unpredictable. In other words, if the program is successful, nothing will happen, as well as in case of unsuccessful program until an unpredictable outbreak. This will give a false feeling assessment of the risk, and might lead to the relaxation of the implementation of biosecurity practices in the first case, or to the feelings of uselessness of the interventions in the latter which, in turn, could increase the probability of disease introduction or transmission [4].”

Author Response

I want to thank the reviewes for the revision of the manuscript. We tried to improve the paper following the suggestions. To complete the material, we are uploading the original version of the BEAT checklist as supplementary material.

The text has been checked for grammar errors and soma sentences have been rewritten.

Details in the file in attachment.

Reviewer 2 Report

Dear authors,

I have read your manuscript with much attention as it discusses an increasingly important subject (biosecurity) in relation to animal-based biomarkers.

Overall I believe the paper is well written, containing valuable information and well-described discussion points. 

The manuscript should be thoroughly rechecked as there are some grammatical and spelling errors and inconsistencies in the references.

Specific comments:

-L55-57: please change this sentence to improve readability

-Table 1: the objectives are sometimes very vague. Maybe the full list of questions/points that were checked on the farms could be added as an attachment.
Also, be consistent with the wording of the different zones (or use the colors or use the "names"; i.e. use of the names throughout the table and the use of "orange zone" within the objectives o the "professional zone". 

- L293: significance level differs from that in the table 

- Table 4: "+-" missing at professional zone score for cluster D

- Lines370 and following: were the measures that improved the most (e.g. manure management) also the measures that were often included in the tailor-made biosecurity plans? We have no information on the latter at the moment. It is unclear if and how many of the measures included in the farm-specific plan were implemented. This would give a different interpretation of the measures that showed high improvement levels.

-L399: roughness?

-L408-412: then how do you explain the overall high initial score of the professional zone?

Author Response

(The authors gave the same response as above.)
